# Comparing Point-of-Care Technology to ELISA Testing for Infliximab and Adalimumab Levels in Adult Inflammatory Bowel Disease Patients: A Prospective Pilot Study

**DOI:** 10.3390/diagnostics14192140

**Published:** 2024-09-26

**Authors:** Erica Bonazzi, Daria Maniero, Greta Lorenzon, Luisa Bertin, Kurtis Bray, Bayda Bahur, Brigida Barberio, Fabiana Zingone, Edoardo Vincenzo Savarino

**Affiliations:** 1Department of Surgery, Oncology and Gastroenterology, University of Padua, 35124 Padua, Italy; erica.bonazzi@unipd.it (E.B.); dariamaniero@gmail.com (D.M.); greta.lorenzon@unipd.it (G.L.); luisa.bertin.1@studenti.unipd.it (L.B.); fabiana.zingone@unipd.it (F.Z.); 2Gastroenterology Unit, Azienda Ospedale—Università Padova, 35128 Padua, Italy; brigida.barberio@gmail.com; 3ProciseDx Inc., 9449 Carroll Park Drive, San Diego, CA 92121, USA; kurt.bray@procisedx.com (K.B.); bayda.bahur@procisedx.com (B.B.)

**Keywords:** point-of-care technology test, inflammatory bowel diseases, therapeutic drug monitoring

## Abstract

**Introduction:** Therapeutic drug monitoring (TDM) has proven to be a valuable strategy for optimizing biologic therapies, among which are anti-tumor necrosis factor (anti-TNF) treatments in inflammatory bowel disease (IBD). In particular, reactive TDM has been shown to manage treatment failures more cost-effectively than empirical dose adjustments for anti-TNF drugs. However, several challenges currently impede the widespread adoption of TDM in clinical practice, particularly addressing the delay between sample collection and result availability. To overcome this limitation, the use of point-of-care technology tests (POCTs) is a potential solution. Point-of-care technology tests are medical diagnostic tests performed at the site of patient care to provide immediate results, allowing for quicker decision-making and treatment. The current standard of care (SOC) for drug level measurement relies on the enzyme-linked immunosorbent assay (ELISA), a method that is time-consuming and requires specialized personnel. This study aims to evaluate a novel, user-friendly, and efficient POCT method (ProciseDx Inc.) and compare its performance with the SOC ELISA in assessing infliximab and adalimumab levels in blood samples from IBD patients. **Methods:** In this prospective, single-center study, we collected blood samples from IBD patients, both CD and UC, receiving infliximab (87 IBD patients; 50% UC and 50% CD) or adalimumab (60 patients; 14% UC and 48% CD) and we analyzed the blood’s drugs levels using both the ProciseDx Analyzer POC and the SOC ELISA. We examined the correlation between the two methods using statistical analyses, including the Deming regression test. Additionally, we assessed the ease of use, turnaround time, and overall practicality of the POCT in a clinical setting. **Results:** The ProciseDx test demonstrated a strong correlation with the SOC ELISA for measuring both infliximab and adalimumab levels. In particular, the overall correlation between the ProciseDx POCT and the ELISA assessments showed an r coefficient of 0.83 with an R squared value of 0.691 (95% CI 0.717–0.902) for IFX measurements, and an r coefficient of 0.85 with an R squared value of 0.739 (95% CI 0.720–0.930). **Conclusions:** the ProciseDx POC test offers significantly faster turnaround times and is more straightforward to use, making it a viable alternative for routine clinical monitoring. Despite its promising potential, further refinement and validation of the ProciseDx test are necessary to ensure its effectiveness across diverse patient populations and clinical settings. Future research should focus on optimizing the POC tests’ performance and evaluating its long-term impact on IBD management.

## 1. Introduction

Inflammatory bowel disease (IBD) encompasses a group of chronic, relapsing, and remitting inflammatory conditions of the gastrointestinal tract, primarily including Crohn’s Disease (CD) and Ulcerative Colitis (UC) [1,2,3,4,5]. Crohn’s disease can affect any part of the gastrointestinal tract, but it is most commonly localized in the ileum and the first part of the colon [6]. It is marked by transmural inflammation, which can lead to complications, such as strictures, fistulas, and abscesses. On the other hand, Ulcerative Colitis is confined to colon and rectum, with inflammation normally restricted to the mucosal layer [7]. The prevalence of IBD is increasing globally, with significant impacts on patients’ quality of life and healthcare systems [8,9,10]. These diseases are characterized by an aberrant immune response causing intestinal inflammation and ulceration of the digestive tract lining [11]. While the exact etiology of IBD remains unknown, it is believed to result from a complex interplay between genetic susceptibility, environmental factors, intestinal microbiota and immune system dysregulation [12]. A range of different symptoms is often experienced by IBD patients, including abdominal pain, diarrhea, rectal bleeding, weight loss, and fatigue. The chronic nature of the disease, along with its unpredictable flares and remissions, poses substantial challenges in patients’ management and treatment. Treatment strategies for IBD aim to induce and maintain remission, improve quality of life, and minimize complications. Therapeutic options include aminosalicylates, corticosteroids, immunomodulators, small molecules, and biologics such as inhibitors of tumor necrosis factor (TNF), anti Il-12/23 antibodies, or anti-integrin antibodies [4,13]. Tumor necrosis factor alpha (TNF-α) is a member of a large family of cytokines that play important roles in inflammation, apoptosis, proliferation, invasion, etc. The overexpression of these cytokines can cause chronic inflammation, leading to autoimmune diseases and tissue damage [14]. The mechanism of action of anti-TNF agents works by binding to the cytokine, preventing it from interacting with its receptors on cell surfaces [15]. This blockade inhibits further downstream inflammatory pathways and reduces the recruitment of inflammatory cells, as well as the release of other pro-inflammatory cytokines. As a consequence, intestinal inflammation is reduced, and this helps mucosal healing [16]. The introduction of anti-tumor necrosis factor (TNF) alpha agents has significantly transformed the prognosis for patients with inflammatory bowel disease (IBD), resulting in higher remission rates and an enhanced quality of life [17,18]. Infliximab (IFX) and adalimumab (ADL) are anti-TNF-α monoclonal antibodies that exert therapeutic effects by inhibiting TNF-α-associated inflammatory responses. Infliximab is administered by intravenous infusion, while adalimumab is injected subcutaneously. Both have shown substantial efficacy in clinical trials and real-world practice in both the induction and maintenance of remission and improving patient’s health-related quality of life [17,19,20,21,22,23,24]. In particular, a study from Kamal and colleagues, aiming to evaluate the efficacy and safety of infliximab and adalimumab in IBD patients, reported that both drugs have a good safety profile and deliver a beneficial clinical and laboratory response in moderate–severe IBD patients [25]. However, several studies reported that a substantial portion of patients were either non-responsive or lost response to the drug during maintenance therapy [26,27].

Therapeutic drug monitoring (TDM) is defined as measuring patients’ response (e.g., drug concentrations, ADA biomarkers, or clinical outcomes) and consequently adjusting the therapy (e.g., the dose or the frequency). The therapeutic drug monitoring of biologic medications for IBD has been introduced in order to optimize treatment for IBD patients who do not respond to standard doses of a specific drug [28,29]. TDM can be used in different ways depending on the situation. Reactive TDM is addressed to investigate treatment failure during the maintenance phase. It involves the evaluation of drug levels and anti-drug antibodies (ADAs) in response to primary non-response or secondary loss of response to therapy. Proactive TDM, on the other hand, is used to potentially prevent treatment failure, and it involves regular monitoring of drug trough concentrations to adjust doses and maintain optimal drug levels. Reactive TDM has been shown to manage treatment failures more cost-effectively than empirical dose adjustments for the anti-TNF drug infliximab [19,27]. Traditional methods like the Enzyme-Linked Immunosorbent Assay (ELISA) are commonly used for therapeutic drug monitoring in blood. However, these techniques demand specialized laboratory facilities and involve lengthy analysis times. This can be particularly challenging in TDM, where timely results are crucial for decision-making in clinical practice. For this reason, the development of point-of-care technology tests (POCTs) presents a promising alternative, offering the potential for faster, more convenient, and accessible monitoring. Although tests can vary based on the method of detection, the type of sample used, and the technology employed, they need to have some characteristics, such as the ease of use and the rapidity in conducting the test and obtaining the results, to be considered suitable for efficiently monitoring therapeutic drug concentrations in patients [30]. The implementation and use of these devices can significantly enhance patient care by providing rapid results followed by faster medical decisions in clinical settings [31,32]. Clinically validated point-of-care technology assays are available for quantifying infliximab and adalimumab levels in serum [33,34]. As the number of such tests increases, it is crucial to assess their quality and accuracy, particularly in comparison with traditional laboratory measurements of drug concentrations, to identify any potential discrepancies. In this context, our study aims to evaluate the performance of a novel POC device for measuring IFX and ADL levels in blood samples from IBD patients, comparing its accuracy, reliability and ease of use with the established ELISA method. These findings may improve patient management by providing timely information to guide treatment decisions in clinical settings.

## 2. Materials and Methods

### 2.1. Study Design

This was a prospective, single-center study aiming to evaluate the performance of a rapid POCT (ProciseDx Inc., San Diego, CA, USA) for quantifying blood levels of IFX (Procise IFX) and ADL (Procise ADL) in IBD patients in comparison to the ELISA reference techniques (Grifol’s Promonitor ELISA test for IFX; Barcelona, Spain). The study was conducted at the Gastroenterology Department of the Padua University Hospital (Italy). All data were anonymized and grouped together for analysis. This study was conducted in full conformity with appropriate local laws and regulations and the Declaration of Helsinki, and the medical ethical committee reviewed the study protocol and approved the study (CESC CODE: 3312/AO/14).

From October 2020 to January 2021, we recruited consecutive patients aged 18 years or older from the Gastroenterology Unit at Padua University Hospital. These patients had been diagnosed with inflammatory bowel disease (IBD), specifically CD or UC, in accordance with the guidelines set forth by the European Crohn’s and Colitis Organisation (ECCO) [35]. The diagnosis of IBD was confirmed through a combination of clinical evaluations, endoscopic procedures, and histopathological examination. Additionally, eligible patients were required to be currently undergoing treatment with either IFX or ADL from at least 12 months. Prior to participation, all patients provided written informed consent, ensuring they were fully aware of the study procedures and potential risks. Several conditions disqualified patients from participating in the study. The patients who were unable or unwilling to adhere to the study protocol or follow-up requirements were not included. A total of 20 µL of capillary whole blood was collected by finger prick and 1.5–2.0 mL of matched venous blood was required to complete all testing [36].

### 2.2. Determination of Infliximab and Adalimumab Levels with the POC Test

For each patient, IFX and ADL levels from capillary whole blood, collected by finger prick with a lancet and collected in a fixed volume pipette (20 µL) (ProciseDx), were evaluated. The capillary blood sample was dispensed into an IFX or ADL reaction cartridge (ProciseDx), which contained lyophilized reagent beads, specific for the measurements. The lot number of the kits used were 20403 and 30219 for infliximab quantification and 20101 for adalimumab quantification. The expiration date was 1 February 2022 for kit 20403, 1 May 2023 for kit 30219, and 1 May 2022 for kit 20101. After the addition of 1.5 mL tris-buffered saline (TBS) to the cartridge, this was closed, inverted five times to allow mixing and the dissolution of the reagent beads, and then placed in the POCT device for analysis [37]. Concurrently, a serum sample from venous blood was collected to carry out the ELISA tests (range 0.035–14.4 µg/mL and 0.024–12 µg/mL for IFX and ADL, respectively). The ProciseDx devices use time-resolved fluorescence resonance energy transfer (TR-FRET) technology to determine IFX and ADL concentrations in capillary blood samples. In particular, the Lumiphore^TM^ technology used allows for a fluorescent emission that is much longer than biologic materials. When both acceptor and donor are bound to the same target protein, the acceptor signal emits light that can be read longer after the biological fluorescent background has passed. The lower and upper limits of the assays were 1.7 µg/mL–77.2 µg/mL for IFX POCT and 1.3 µg/mL–51.5 µg/mL for ADL POCT.

### 2.3. Feasibility

To evaluate the feasibility of the POCT, we aimed to assess its practicality, ease of use in a clinical setting, and timing efficiency. This included determining the success rate of blood collection using the POCT and identifying any instances where additional samples were required. Specifically, we monitored the number of patients for whom the initial finger-prick blood collection was sufficient for analysis, and recorded any cases that necessitated repetition of the blood collection due to inadequate sample volume or other issues. We measured the time from blood collection to obtaining the test results in order to determine the speed of the POCT and compare it to the serum ELISA method. This study was designed as a pilot study, primarily intended to explore feasibility, gather preliminary data, and identify trends that could inform future, larger-scale studies. As such, we did not perform a formal sample size calculation based on statistical power. The selection of patients was driven by practical considerations, including the availability of eligible patients during study period and ensuring a sufficient number of participants to provide meaningful exploratory insights.

### 2.4. Statistical Analysis

Descriptive analyses on the demographic and clinical characteristics of the study sample were calculated to summarize the key characteristics of the dataset. Measures of central tendency, including the mean, were used to provide insight into the typical values of the variables under study. Measures of dispersion, such as the standard deviation, were employed to assess the variability within the data. All statistical analyses were conducted using the R software package version 4.0.3. The results provide a comprehensive overview of the data, facilitating the identification of patterns and potential outliers that may influence subsequential analyses.

In order to made a comparison between the measurements performed with the Procise assays and the ELISA test, a Deming regression test was employed, which accounts for measurements errors in both variables. Indeed, this method was chosen due to its ability to provide more accurate parameter estimates compared to ordinary least-squares regression when both variables are subjected to error. All analyses were performed using R version 4.0.3.

## 3. Results

### 3.1. Feasibility and Correlation of Procise IFX Point-of-Care Technology Test with ELISA for Assessing Infliximab Levels in Blood Samples from IBD Patients

#### 3.1.1. Population Description

Eighty-seven patients were enrolled in the study, with a mean age of 44 ± 1.7 years, comprising 63% men and 37% women. Of them, 49% were diagnosed with CD and 51% with UC. The range of disease duration was between 12 and 456 months, with a mean of 134 months. The patients started biologic treatment at least 12 months before, with a mean duration of the treatment of 54 months. All demographic and clinical characteristics of the study population are reported in Table 1.

#### 3.1.2. Feasibility

The assessment using the Procise IFX POCT was feasible for each patient, and only three instances required a repeat finger-prick blood collection.

The time from blood collection to obtaining the results was approximately 3 ± 0.5 min with the Procise IFX POCT, whereas the serum ELISA analysis necessitated the collection of at least 40 samples, typically taking around three weeks at our center, plus an additional 3 h to perform the analysis.

#### 3.1.3. Correlation between Procise IFX POCT and ELISA Measurements

Among the patients, 39 (59% males; mean age of 44 ± 16 years) had trough levels of infliximab assessed by the Procise IFX, which were either lower than 1.7 µg/mL or greater than 14.4 µg/mL, consistent with the ELISA results. For this reason, these patients were excluded from the Deming regression analysis.

For the remaining 48 patients (67% men and 33% women), there was a strong correlation between the results of the two tests (Figure 1). The overall correlation between the Procise IFX POCT and the ELISA assessments showed an r coefficient of 0.83 and an R squared value of 0.691 (95% CI 0.717–0.902). 

### 3.2. Feasibility and Correlation of Procise ADL Point-of-Care Technology Test with ELISA for Assessing Adalimumab Levels in IBD Patients

#### 3.2.1. Population Description

Sixty patients were enrolled in the study, comprising 67% men and 33% women with a mean age of 43 ± 14 years. Among them, 80% were diagnosed with CD and 20% with UC. The range of disease duration was between 12 and 516 months, with a mean of 140 months. The patients started biologic treatment at least 12 months before, with a mean duration of the treatment of 19 months. All demographic and clinical characteristics of the study population are reported in Table 2. 

#### 3.2.2. Feasibility

The Procise ADL POCT assessment was feasible for all patients, with a rapid turnaround time of 3 ± 0.2 min. In contrast, serum ELISA analysis required the collection of at least 40 samples to perform batch analysis, typically taking around three weeks at our center, plus an additional 3 h for processing.

#### 3.2.3. Correlation between Procise ADL Assay POCT and ELISA Measurements 

For 30 patients (63% men and 37% women), the Procise ADL indicated trough levels of adalimumab either below 1.3 µg/mL or above 12 µg/mL, which was consistent with the ELISA results. For this reason, these patients were excluded from the Deming regression analysis.

For the remaining thirty patients (70% men and 30% women), there was a strong correlation between the Procise ADL POCT and the ELISA assessments (Figure 2), with an r coefficient of 0.85 and an R squared value of 0.739 (95% CI 0.720–0.930).

## 4. Discussion

Therapeutic drug monitoring is crucial in clinical practice to ensure that drug concentrations remain within a therapeutic range, thus optimizing efficacy while minimizing toxicity and adverse effects [38]. Indeed, the exposition of IBD patients to sub-therapeutic drug concentrations may represent the reason for loss of response, together with the formation of antibodies against the drug, and this condition represents the most common mechanism of low or undetectable drug concentration [39]. In particular, research has shown that treating patients following the “one size fits all” principles is no longer the best option in IBD treatment, since not all patients respond the same way to a certain dosing regimen [40]. Altogether, these observations emphasize the need for adequate, more individualized and personalized dosing therapy, in which therapeutic drug monitoring plays an important role. TDM may be considered one of the first tools of precision medicine, with the aim of individualizing the dose of drugs based on patient characteristics both a priori (demographics, pharmacogenetics, clinical information, etc.) and/or a posteriori by the measurement of drugs in blood, and/or of biomarkers in blood or other biological fluids [41]. In order to quickly monitor and develop more personalized treatment for IBD patients, novel point-of-care technology testing for TDM has been developed in recent years, which is also used for monitoring oral small-molecule and biologic drug concentrations [42]. In general, the important characteristics for effective POC testing in therapeutic drug monitoring are accuracy and precision, sensitivity and specificity, ease of use, and rapidity in execution and in obtaining results. A recently developed example of a next-generation point-of-care technology test for the therapeutic drug monitoring of biologic drugs is “the whole blood in—result out” device designed by Ordutoski and Da Dosso, consisting of a self-powered microfluidic chip for the quantification of ADL levels in blood samples from IBD patients [30]. Using this test, the required analysis time needed for ADL quantification in whole capillary blood samples is around 30 min. Compared to this next-generation POC test, our ProciseDx device demonstrated a sixfold shorter turnaround time, with less than 5 min for both IFX and ADL quantification, thus appearing more suitable for daily clinical practice use. In our study, we evaluated the accuracy, ease of use, and rapidity of a novel point-of-care test, assessing infliximab (ProciseDx IFX) and adalimumab (ProciseDx ADL) levels in blood samples from IBD patients. We recruited patients suffering from both Crohn’s Disease and Ulcerative Colitis and receiving a biologic treatment with either infliximab or adalimumab. Firstly, the point-of-care test ProciseDx proved to be feasible for use in a clinical setting, with a rapid turnaround time of just 3 ± 0.2 min for both IFX and ADL quantification. In particular, the time required for running the analysis and obtaining the results was significantly less compared to the time required for the traditional ELISA method (currently the standard of care for therapeutic drug monitoring for IFX and ADL) [43], which typically takes several days, thus highlighting the potential of the ProciseDx Analyzer as a rapid and reliable alternative for TDM in patients with IBD. Our study demonstrated that the ProciseDx IFX and ADL Analyzer has a strong correlation with the established ELISA method for standard of care, with an r coefficient of 0.83 and an R squared value of 0.691 for infliximab quantification and an r coefficient of 0.85 and an R squared value of 0.739 for adalimumab quantification, both measured in blood samples from IBD patients. The correlation levels in our study between both assays were comparable to those reported in previous studies where different POC devices and ELISA standard-of-care tests were compared for the quantification of IFX and ADL in blood samples from IBD patients [32,44,45,46]. An advantage of the ProciseDx POC test is that, in contrast to the standard-of-care test and other POC tests, which require serum samples [47], this device directly analyzes capillary whole blood using finger-prick samples. This allows results to be obtained immediately and individually for each patient, without the need for batch sampling and serum separation before processing and analysis, as is required for ELISA tests. 

The limitations of this study should be recognized. Due to the small sample size included in the study (48 patients for IFX level monitoring and 30 patients for ADL level monitoring) and the single-center aspect, it is challenging to generalize these results to a broader population. Indeed, while the ProciseDx Analyzer POCT performed well, further research is needed to assess its performance in various clinical settings and with different patient populations. Technical limitations related to sample collection and assay accuracy at extreme drug concentrations also warrant further investigation.

The implementation and use of the ProciseDx Analyzer POCTs for clinical practice could help IBD management by providing quicker results and enabling quicker therapeutic decisions. In IBD, medical decisions and timely adjustments to treatment are crucial for maintaining remission and preventing complications and flares [31]. Moreover, the ability to quickly measure drug levels in blood with the Procise IFX and ADL POCT can facilitate more responsive and personalized treatment strategies [48]. Indeed, it will be crucial to explore the impact of this device on clinical outcomes in order to determine whether its rapid turnaround time and strong correlation with the standard ELISA method translate into improved disease management, reduced hospitalizations, fewer disease complications, and enhanced patient quality of life. Multi-center trials would provide robust data and enhance the validation of our findings across various clinical settings. Additionally, longitudinal studies tracking patients over extended periods could offer insights into the long-term benefits and reliability of the POC system. Comparative research with other POC systems could identify its relative advantages and specific areas for improvement.

## 5. Conclusions

By demonstrating the efficacy and accuracy of the ProciseDx Analyzer POC in measuring IFX and ADL levels in blood samples from IBD patients, this study adds valuable evidence supporting its use in clinical practice, potentially leading to better therapy outcomes through a faster, personalized, and reliable drug monitoring approach. The validation of these results in a broader context, including multi-center studies as well as monitoring drug concentrations over extended periods, are needed before the implementation of these tests in daily clinical practice. Moreover, it would be useful and interesting to assess the impact of this device on clinical outcomes, such as disease management, patients’ quality of life, and disease complications. 

## Figures and Tables

**Figure 1 diagnostics-14-02140-f001:**
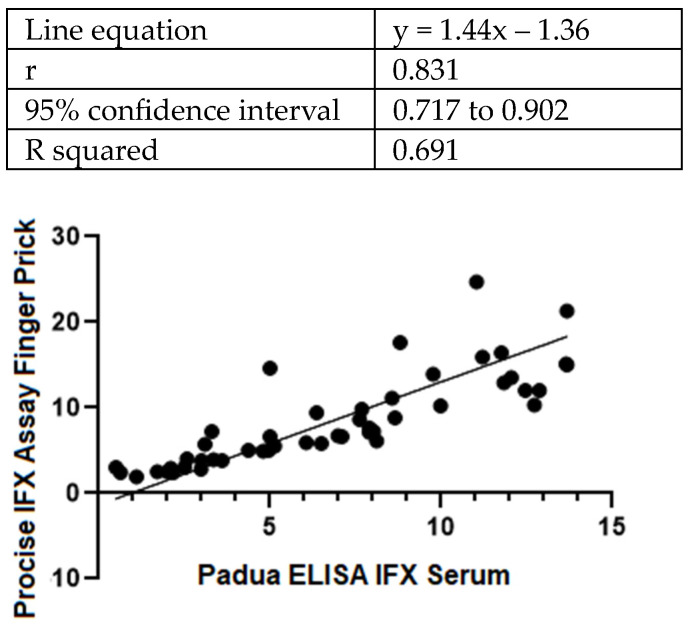
Deming regression between results from the Procise IFX POCT and the ELISA measurements of infliximab levels in blood samples from IBD patients (N = 48).

**Figure 2 diagnostics-14-02140-f002:**
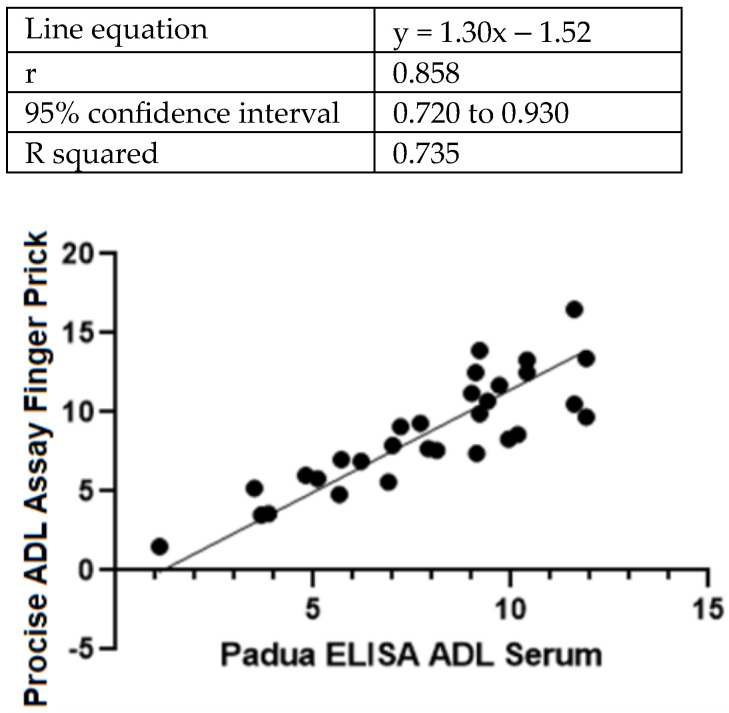
Deming regression between results from the Procise ADL POCT and the ELISA measurements of adalimumab levels in blood samples from IBD patients (N = 30).

**Table 1 diagnostics-14-02140-t001:** Demographic and clinical characteristics of the studied sample (IFX).

Number of patients, *n*	87
Type of disease	
UC, *n*	44
CD, *n*	43
Mean age, *n* (s.d.)	44 (1.7)
Sex (male/female), *n*	52/35
Mean disease duration (months), *n* (range)	134 (12–456)
Mean duration of biologic treatment (months), *n* (range)	54 (12–192)

**Table 2 diagnostics-14-02140-t002:** Demographic and clinical characteristics of the studied sample (ADL).

Number of patients	60
Type of disease	
UC, *n*	12
CD, *n*	48
Mean age, *n* (s.d.)	43.5 (1.8)
Sex (male/female), *n*	38/22
Mean disease duration (months), *n* (range)	140 (12–516)
Mean duration of biologic treatment (months), *n* (range)	19 (12–168)

## Data Availability

The data supporting the findings of this study are available upon reasonable request from the corresponding author.

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
