# Peer review of "Comparing Point-of-Care Technology to ELISA Testing for Infliximab and Adalimumab Levels in Adult Inflammatory Bowel Disease Patients: A Prospective Pilot Study"

_diagnostics, 2024, doi:10.3390/diagnostics14192140_

Round 1

Reviewer 1 Report

Comments and Suggestions for Authors

The overall impression of the study raises questions for me, at least the statistical analysis is at a very low level, for example, it is not deciphered what is +- or normal distribution is checked, no p-value is found anywhere, correlation analysis is used to compare the two methods, but in reality it should have been used ICC coefficient to compare agreement.

Author Response

Comments and Suggestions for Authors

The overall impression of the study raises questions for me, at least the statistical analysis is at a very low level, for example, it is not deciphered what is +- or normal distribution is checked, no p-value is found anywhere, correlation analysis is used to compare the two methods, but in reality it should have been used ICC coefficient to compare agreement.

Author response: we thank the reviewer for this important feedback regarding the statistical analysis in our study. We agree with you that the ICC coefficient can be used to compare agreement between measurements. However, in our study we chose the Deming regression analysis to take into account for measurement errors in both variables, making it suitable for comparing two methods. The aim of our study was to evaluate relationship between two variables, measured with two different methods. For this analysis, Deming regression is normally used with the goal to compare one measurement to another one (e.g comparing a new method with a standard of care) [1], [2].

Submission Date

08 August 2024

Date of this review

30 Aug 2024 11:39:23

[1]       K. Linnet, «Necessary Sample Size for Method Comparison Studies Based on Regression Analysis», Clinical Chemistry, vol. 45, fasc. 6, pp. 882–894, giu. 1999, doi: 10.1093/clinchem/45.6.882.

[2]       S. Huang, Method comparison and method calibration applicable to forest measurements and model predictions. Edmonton, Alberta, Canada: Alberta Agriculture and Forestry, 2019.

Reviewer 2 Report

Comments and Suggestions for Authors

Authors have evaluated the POC testing vs Standard care testing for IFX and Adalimumab testing. 

Authors have done a well constructed study evaluating the role of POC testing. I have few queries and comment.

Although they have evaluated this in both population, extreme values especially low values were excluded and is more clinically relevant. 

Authors have not discussed anti drug antibodies, atleast in USA the durg levels has Anti drug antibodies levels which is important, especially in the context of low values and may change management. 

Author Response

Comments and Suggestions for Authors

Authors have evaluated the POC testing vs Standard care testing for IFX and Adalimumab testing. 

Authors have done a well constructed study evaluating the role of POC testing. I have few queries and comment.

Author response: we thank the reviewer for the positive feedback and for acknowledging the structure of our study. We greatly appreciate the interest and the opportunity to address the queries and comments below.

Although they have evaluated this in both population, extreme values especially low values were excluded and is more clinically relevant. 

Author response: we thank the reviewer for this comment. Although we agree that low values can be clinically relevant, these values had to be excluded from our study, as well as too high values, because they fell outside the indicated confidence range of the device.

Authors have not discussed anti drug antibodies, atleast in USA the durg levels has Anti drug antibodies levels which is important, especially in the context of low values and may change management. 

Author response: We thank the reviewer for the valuable comment. We fully agree that anti-drug antibodies play a critical role in the management of therapies, particularly in relation to drug levels and their impact on therapeutic efficacy. However, the primary aim of our study was to correlate point-of-care (POC) technology results with ELISA results to assess the reliability of POC testing compared to the current standard of care method. The scope of our research was specifically focused on evaluating the accuracy and feasibility of drug level measurements between these two techniques. We acknowledge that for comprehensive therapy management in clinical settings it is essential to incorporate other tests, including those that evaluate anti-drug antibodies, to gain a complete understanding of treatment response and drug efficacy. Future studies may benefit from integrating ADA testing alongside drug level monitoring to provide a more holistic approach to therapeutic management.

Submission Date

08 August 2024

Date of this review

11 Sep 2024 04:12:40

Reviewer 3 Report

Comments and Suggestions for Authors

This is a pilot comparison of two technologies designed to measure blood concentrations of two TNF-blocking agents used in Crohn and ulcerative colitis, adalimumab and infliximab.

Title: shorten the title by changing "Comparative Assessment" to "Comparing" and "current Standard of Care" by removing parentheses around ELISA. Spell out IBD. Add Adult. Remove "Quantification in Blood Samples." Proposed new title: "Comparing Point-of-Care versus ELISA Testing for Infliximab and Adalimumab Levels from Adult Inflammatory Bowel Disease Patients: A Prospective Pilot Study."

Abstract: Include the number of patients tested and the % with Crohn and UC.

Keywords: To be more consistent with MeSH terms, consider changing "point-of-care test" to "point of care technology" and "monitoring drug levels" to "therapeutic drug monitoring." Add the drug names individually or the drug class, "tumour necrosis factor alpha." Alphabetize them.

Ln 124-125: For the specific test units, include the lot numbers and expiration dates. 

Ln 132: how was the sample size determined? Why were 87 patients selected?

Lns 189 and 221: You tested a sample, so change population to sample.

Ln 332: The study was partially funded and two of the authors are employed by Procise. The ethics statement needs to include a sentence about how the company was involved in the statistical analyses and writing and interpretation of the results.

Ln 244: There are other POCT available in Italy, including Promonitor Quick and Grifols POC. Why did you choose Procise?

Ln: 295: Limitations are well explained.

Ln 326: change etc. to among others.

Ln 347-359: The conflict of interest should state that Kurtis Bray and Bayda Bahur are employees of Procise

References are in MDPI style.

Thank you for the opportunity to review your work.

Author Response

Comments and Suggestions for Authors

This is a pilot comparison of two technologies designed to measure blood concentrations of two TNF-blocking agents used in Crohn and ulcerative colitis, adalimumab and infliximab.

Title: shorten the title by changing "Comparative Assessment" to "Comparing" and "current Standard of Care" by removing parentheses around ELISA. Spell out IBD. Add Adult. Remove "Quantification in Blood Samples." Proposed new title: "Comparing Point-of-Care versus ELISA Testing for Infliximab and Adalimumab Levels from Adult Inflammatory Bowel Disease Patients: A Prospective Pilot Study."

Author response: we thank the reviewer for this suggestion. The title has been changed in the revised manuscript as proposed.

Abstract: Include the number of patients tested and the % with Crohn and UC.

Author response: we thank the reviewer for this suggestion. We added the requested data in the abstract of the revised manuscript.

Keywords: To be more consistent with MeSH terms, consider changing "point-of-care test" to "point of care technology" and "monitoring drug levels" to "therapeutic drug monitoring." Add the drug names individually or the drug class, "tumour necrosis factor alpha." Alphabetize them.

Author response: we thank the reviewer for this comment. We edited the revised manuscript as proposed.

Ln 124-125: For the specific test units, include the lot numbers and expiration dates. 

Author response: We thank the reviewer for this important comment. We have now edited the revised manuscript accordingly, adding the lot number and expiration dates of the kits which have been used for the quantification of adalimumab and infliximab levels.

Ln 132: how was the sample size determined? Why were 87 patients selected?

Author response: we thank the reviewer for this important feedback. However, this was considered as a pilot study, intended to explore feasibility and gather preliminary data that could inform future, larger-scale studies. For this reason, we did not perform a formal sample size calculation based on statistical power and the selection of patients was driven by practical considerations. We have now mentioned this point in the revised manuscript

Lns 189 and 221: You tested a sample, so change population to sample.

Author response: we thank the reviewer for this clarification. We have now changed the revised manuscript accordingly.

Ln 332: The study was partially funded and two of the authors are employed by Procise. The ethics statement needs to include a sentence about how the company was involved in the statistical analyses and writing and interpretation of the results.

Author response: we thank the reviewer for this comment. We now added the contributions of the two authors employed by Procise in the “Author Contributions” statement in the revised manuscript

Ln 244: There are other POCT available in Italy, including Promonitor Quick and Grifols POC. Why did you choose Procise?

Author response: we thank the reviewer for this question. A previous study from our team, already tested the Promonitor Quick POC and we discussed these results in a scientific article [1]. The Procise POC was selected to assess the results of another device.

Ln: 295: Limitations are well explained.

Author response: we thank the reviewer for this positive feedback

Ln 326: change etc. to among others.

Author response: we thank the reviewer for this comment. We now edited the revised manuscript accordingly

Ln 347-359: The conflict of interest should state that Kurtis Bray and Bayda Bahur are employees of Procise.

Author response: we thank the reviewer for this comment. We now added the contributions of the two authors employed by Procise in the “Author Contributions” statement in the revised manuscript

References are in MDPI style.

Thank you for the opportunity to review your work.

Submission Date

08 August 2024

Date of this review

28 Aug 2024 20:36:22

[1]       S. Facchin et al., «Rapid point-of-care anti-infliximab antibodies detection in clinical practice: comparison with ELISA and potential for improving therapeutic drug monitoring in IBD patients», Therap Adv Gastroenterol, vol. 14, p. 1756284821999902, mar. 2021, doi: 10.1177/1756284821999902.

Round 2

Reviewer 1 Report

Comments and Suggestions for Authors

The statistical analysis was completely meaningless, only the correlation was calculated and the univariate regression analysis was performed, and the authors also use the standard deviation together with the median (Table 1), which shows the complete absence of knowledge of medical statistics.
